

# Importance of Ammonia Gas-Particle Conversion Ratio in Haze Formation in the Rural Agricultural Environment

Jian Xu[1], Jia Chen[1], Na Zhao[1], Guochen Wang[1], Guangyuan Yu[1], Hao Li[1], Juntao Huo[3], Yanfen Lin[3], Qingyan Fu[3], Hongyu Guo[4], Congrui Deng[1,2], Shan-Hu Lee[5], Jianmin Chen[1], Kan Huang[1,2,6]*

[1]Shanghai Key Laboratory of Atmospheric Particle Pollution and Prevention (LAP[3]), Department of Environmental Science and Engineering, Fudan University, Shanghai 200433, People's Republic of China
[2]Institute of Eco-Chongming (IEC), 20 Cuiniao Road, Chenjiazhen, Shanghai 202162, People's Republic of China
[3]Shanghai Environmental Monitoring Center, Shanghai 200235, People's Republic of China
[4]Cooperative Institute for Research in Environmental Sciences and Department of Chemistry, University of Colorado, Boulder, Colorado 80309, United States
[5]Department of Atmospheric & Earth Science, University of Alabama in Huntsville, Huntsville, Alabama 35758, United States
[6]Institute of Atmospheric Sciences, Fudan University, Shanghai 200433, People's Republic of China

*Correspondence to*: Kan Huang (huangkan@fudan.edu.cn)

**Abstract.** Ammonia in the atmosphere is essential for the formation of fine particles that impact air quality and climate. Despite extensive prior research to disentangle the relationship between ammonia and haze pollution, the role of ammonia in haze formation in the high ammonia emitted regions is still not well understood. Aiming to better understand secondary inorganic aerosol (SNA) formation mechanisms under high ammonia conditions, one-year hourly measurement of water-soluble inorganic species (gas and particle) was conducted in a rural supersite in Shanghai. Exceedingly high levels of agricultural ammonia, constantly around 30 ug m$^{-3}$, were observed. We find that ammonia gas-particle conversion ratio (ACR), as opposed to ammonia concentrations, plays a critical role in SNA formation during the haze period. By assessing the effects of various parameters, including temperature (T), aerosol water content (AWC), aerosol pH, and activity coefficient, it seems that AWC plays predominant regulating roles for ACR. We propose a self-amplifying feedback mechanism associated with ACR for the formation of SNA, which is consistent with diurnal variations of ACR, AWC, and SNA. Our results imply that reduction of ammonia emissions alone may not reduce SNA effectively at least in rural agricultural sites in China.

## 1 Introduction

Gas-phase ammonia (NH$_3$) in the environment not only fuel the eutrophication and acidification of ecosystems, but also play key roles in atmospheric chemistry. NH$_3$ has been known to promote new particle formation both in the initial homogeneous nucleation and subsequent growth (Ball et al., 1999;Zhang et al., 2011;Coffman and Hegg, 1995;Kirkby et al., 2011). Prior studies suggest that the SO$_2$ oxidation can be enhanced by the presence of NH$_3$ (Turšič et al., 2004;Wang et al., 2016;Benner et al., 1992). High levels of NH$_3$ can also promote SOA formation (Na et al., 2007;Ortiz-Montalvo et al., 2013). As the main alkaline species in the atmosphere, NH$_3$ are expected to affect the acidity of clouds (Wells et al., 1998), fine particles (Liu et al., 2017;Guo et al., 2017b), and wet deposition (ApSimon et al., 1987) by neutralizing acidic species. The neutralized ammonium (NH$_4$$^+$) exclusively contribute to



aerosol hygroscopicity especially in hazy periods (Liu et al., 2017;Ye et al., 2011). Serving as efficient catalysts for
aerosol aldol condensation, ammonium has also been proved to contribute to radiative forcing (Noziere et al.,
2010;Park et al., 2014). Most importantly, ammonium is among the major secondary inorganic aerosols (i.e., sulfate
$SO_4^{2-}$, nitrate $NO_3^-$, and ammonium $NH_4^+$, denoted as SNA), which typically rivals the organics and can make up
more than 50% of $PM_{2.5}$ mass loadings (Wang et al., 2015a;Sun et al., 2014;Huang et al., 2014;Plautz, 2018;Schiferl
et al., 2014). Despite the significant importance of SNA in hazy periods, its formation mechanism responsible,
particularly the role of $NH_3$, remains highly controversial. Cheng et al. (2016) and Wang et al. (2016), for example,
suggested that the near-neutral acidity, resulting from the $NH_3$ rich atmosphere, is vital for SNA formation. While
Liu et al. (2017) and Guo et al. (2017b) demonstrated that the close to neutral state is unlikely even under conditions
of excess $NH_3$. These findings collectively imply that the fundamental role of $NH_3$ in regulating aerosol acidity is
still ambiguous, thus altering the SNA formation mechanism (Seinfeld and Pandis, 2012).

$NH_3$ emission sources include agricultural practices, on-road vehicles(Chang et al., 2016;Sun et al., 2016) and
biomass burning (Lamarque et al., 2010;Paulot et al., 2017). Recent field measurements and modeling works reveal
that agricultural practices (i.e., animal manure and fertilizer application) contribute to 80-90% of total $NH_3$
emissions in China (Zhang et al., 2018;Kang et al., 2016;Huang et al., 2011). Globally, $NH_3$ emissions are projected
to continue to rise along with increasing demand of chemical fertilizers due to the growing human population
(Erisman et al., 2008;Stewart et al., 2005) and in part because limiting $NH_3$ emissions has not been targeted a
priority in most countries. For example, even though stringent mitigation targets have been set for $SO_2$ and $NO_x$ in
China's 13th Five-Year Plan (2016-2020), slashing $NH_3$ emissions is not yet a prime concern in China. The
sustained increase of $NH_3$ has been observed from the space (Warner et al., 2017) and reported to deflect the
mitigation efforts of $SO_2$ and $NO_x$ emissions in East China (Fu et al., 2017).

Although agricultural $NH_3$ emission has been the subject of extensive research, previous studies have focused on
densely populated or urban areas, where $NH_3$ was mostly "aged" and transformed to $NH_4^+$ downwind (Chang et al.,
2016). Varying in location and time, the typical mass concentrations of $NH_3$ are on the order of several micrograms
61 per cubic meter(Yao et al., 2006;Gong et al., 2013;Robarge et al., 2002;Chang et al., 2016;Phan et al., 2013), with
62 extremely high levels up to more than 20 µg m$^{-3}$ in the rural area of North China Plain(Meng et al., 2018;Shen et al.,
2011;Pan et al., 2018). Numerous studies highlighted the importance of $NH_3$ emissions from agricultural areas
(Meng et al., 2018;Shen et al., 2011;Robarge et al., 2002;Wang et al., 2013;Nowak et al., 2012;Zhang et al.,
2017;Warner et al., 2017), but the gas-particle conversion of agricultural $NH_3$ in rural regions and its subsequent
impact on SNA formation, has scarcely been reported and remains poorly understood.

In this study, we provide observational constraints on the abnormally high agricultural $NH_3$ emission at a rural site.
We report our findings on the influence of $NH_3$ gas-particle conversion ratio on SNA formation and discuss the
decisive factors driving the $NH_3$ gas-particle conversion ratio (ACR).





## 2 Methods

### 2.1 Observation site

Field measurements of gases and fine particles were conducted over the course of a year from March 2017 to February 2018 at the Dongtan Wetland Park (31°32′ N, 121°58′ E; altitude: 12 m a.s.l.), which is approximately 50 km northeast of downtown Shanghai. The sampling site, illustrated in Figure 1, was located on the east side of the Chongming island, which is the largest eco-friendly island in China and the least developed district of Shanghai. The annual mean relative humidity (RH) is 78% ± 19% and the yearly average temperature (T) is 16.3 ± 9.9℃. Although Chongming shares limited industrial and vehicle emissions compared to urban Shanghai, the level of fine particles on this island is slightly higher than the urban site (Figure S1). The overuse of nitrogen fertilizer has long been a large agricultural source of $NH_3$ emissions in China (Fan et al., 2011), with an increasing use especially in East-Central China (Yang and Fang, 2015), where rice/wheat intercropping (similar to those in Chongming) was applied. Based on a 2011 agricultural $NH_3$ emission inventory in Shanghai, Chongming has the largest nitrogen fertilizer consumption among all the districts in Shanghai (Fang et al., 2015). According to the Multi-resolution Emission Inventory for China (MEIC, www.meicmodel.org) in 2016, nearly 94% of $NH_3$ in Chongming came from the agricultural sector, accounting for 14% of the total $NH_3$ emissions in Shanghai. In comparison, Chongming contributes only 6% and 5% of the total $NO_x$ and $SO_2$ emissions in Shanghai, respectively (Table S1). With the most intensive agriculture and 34% of arable farmland area in Shanghai (Wen et al., 2011), atmospheric ammonium aerosols over the Chongming island are mostly of agricultural origin. Therefore, this site is ideal for investigating the role of agricultural emissions of $NH_3$ in haze formation.

### 2.2 Measurements

Water-soluble samples of both gases ($NH_3$, $SO_2$, HCl, $HNO_2$, and $HNO_3$) and particles ($NH_4^+$, $Na^+$, $K^+$, $Ca^{2+}$, $Mg^{2+}$, $Cl^-$, $NO_3^-$, and $SO_4^{2-}$) were measured hourly using MARGA (Monitor for AeRosols and Gases in Air, ADI 2080, Metrohm Applikon B.V., Netherlands). Online sampling was conducted from March 2017 to February 2018 following the description in Kong et al. (2014). Briefly, air was drawn into a $PM_{2.5}$ cyclone inlet with a flow rate of 1 $m^3$ $h^{-1}$ and passed through either a wet rotating denuder (gases) or a steam jet aerosol collector (aerosols). Subsequently, the aqueous samples were analyzed with ion chromatography. Meanwhile, $PM_{2.5}$ and gaseous pollutants ($SO_2$, $NO_2$, $O_3$, and CO) were monitored by co-located instruments. Mass loadings of $PM_{2.5}$ was determined by a Tapered Element Oscillating Microbalance coupled with Filter Dynamic Measurement System (TEOM 1405-F). $SO_2$ mass concentrations were analyzed by Pulsed Fluorescence $SO_2$ Analyzer (Thermo Fisher Scientific, Model 43i). $NO_2$ mass concentrations were analyzed by Chemiluminescence $NO$-$NO_2$-$NO_x$ Analyzer (Thermo Fisher Scientific, Model 42i). $O_3$ mass concentrations were analyzed by UV Photometric Ozone Analyzer (Thermo Fisher Scientific, Model 49i). CO mass concentrations were analyzed by Gas Filter Correlation CO Analyzer (Thermo Fisher Scientific, Model 48i). The QA/QC of these instruments were managed by professional staff in Shanghai Environmental Monitoring Center (SEMC) according to the Technical Guideline of Automatic Stations of Ambient Air Quality in Shanghai (HJ/T193-2005).



### 2.3 ISORROPIA-II modelling

The thermodynamic model ISORROPIA II (Fountoukis and Nenes, 2007) was used to predict the aerosol water content and pH. ISORROPIA was constrained in forward metastable mode by hourly averaged measurements of $Na^+$, $K^+$, $Mg^{2+}$, $Ca^{2+}$, $SO_4^{2-}$, $NH_3$, $NH_4^+$, $HNO_3$, $NO_3^-$, HCl, and $Cl^-$, along with RH and T. The molality based pH was a default output in the model. The model showed a good performance when predicting $NH_3$-$NH_4^+$ partitioning (Figure 2).

### 3 Results and Discussion

### 3.1 $NH_3$ levels and its link to secondary inorganic aerosol

Figure 3 shows that the mean concentration of $NH_3$ at Chongming (CM: $17.0 \pm 4.2$ µg m$^{-3}$) is more than three times higher than an urban site in Shanghai (PD: $2.5 \pm 0.9$ µg m$^{-3}$) and a representative regional transport region (DL: $4.6 \pm 2.0$ µg m$^{-3}$) in the Yangtze River Delta. The level of $NH_3$ at Chongming is even close to that observed inside a typical dairy farm (JS: $19.4 \pm 12.6$ µg m$^{-3}$), which is dominated by livestock emissions. Thus, it is interesting to investigate how the formation of secondary inorganic aerosols is impacted by this abnormally high level of $NH_3$. Figure 4 indicates the response of SNA (sulfate, nitrate, and ammonium) mass concentrations to $NH_3$ is nonlinear. Higher $NH_3$ sometimes correspond to even lower SNA mass concentrations. Statistically, the average SNA concentration in each bin of $NH_3$ doesn't show significant difference. This is at odds with the traditional view that higher concentrations of precursors usually result in elevated inorganic aerosols (Nowak et al., 2010). Although the abundance of SNA is related to the alkaline gaseous precursor (e.g., $NH_3$), the ambient condition (e.g., RH and T), and acid precursors (i.e., $SO_2$ and $NO_x$) whether favor the conversion of precursors into particles or not is equally important, if not higher. For example, the urban areas show higher SNA levels than the rural region while lower $NH_3$ mixing ratio was observed (Wu et al., 2016;Wang et al., 2015b). Previous field measurements suggest that rural $NH_4^+$ levels are more sensitive to acidic gases than to the $NH_3$ availability (Shen et al., 2011;Robarge et al., 2002). Therefore, the level of $NH_3$ concentration is not the determining factor for the formation of secondary inorganic aerosols.

### 3.2 The role of ammonia gas-particle conversion ratio

In this regard, we further investigate the relationship between the ammonia gas-particle conversion ratio (ACR, defined as the molar ratio between particle phase ammonia ($NH_4^+$) and total ammonia ($NH_x = NH_3+NH_4^+$)) and SNA during the haze period. The haze period is defined as hourly averaged $PM_{2.5}$ mass loadings higher than 75 µg m$^{-3}$. As shown in Figure 5, it is obvious that SNA in $PM_{2.5}$ is almost linearly correlated with ACR. Higher ACR results in higher SNA concentrations. In addition, under the same ACR conditions, higher $NH_3$ promotes stronger formation of SNA. Thus, $NH_3$ and ACR collectively determine the haze formation potential. The level of $NH_3$ can be regarded as a proxy of $NH_3$ emission intensity, which is source dependent. As for ACR, it represents the relative abundance of gaseous $NH_3$ and particulate ammonium. The shift between the two phases is controlled by various factors such as the ambient environmental conditions. Previous study shows that elevated RH and acidic gas levels favor the shift





of NH$_3$ towards the particulate phase at an urban site, thereby a lower [NH$_3$]:[NH$_4^+$] ratio was observed (Wei et al.,

2015). In this study, it is also observed that higher ACR values coincide with heightened RH, SO$_2$, and NO$_x$.

Based on the above results, elucidation of the driving factors determining ACR is of great importance to explore the

formation mechanism of haze. Theoretically, ACR is determined by NH$_3$, NH$_4^+$, and the equilibrium between NH$_3$

and NH$_4^+$. Assuming NH$_3$ and NH$_4^+$ are in thermodynamic equilibrium, the following equation can be obtained.

$\mathrm{H^+ + NH_{3(g)} \leftrightarrow NH_4^+}$ (R1)

The equilibrium constant $H_{NH_3}^*$ is equal to the Henry's constant of NH$_3$ divided by the acid dissociation constant for

NH$_4^+$ (Clegg et al., 1998b). $H_{NH_3}^*$ is calculated by the following equation:

$\ln(H_{NH_3}^*) = 25.393 - 10373.6(1/T_r - 1/T) + 4.131(T_r/T - (1 + \ln(T_r/T)))$ (Eq. 1)

here, T$_r$ is the reference temperature of 298. 15 K. ACR can be analytically calculated as detailed in Guo et

al.(2017a) via the following equation:

$\mathrm{ACR} = \dfrac{[NH_4^+]}{[NH_x]} \cong \dfrac{\frac{\gamma_{H^+} 10^{-pH}}{\gamma_{NH_4^+}} H_{NH_3}^* W_i RT \times 0.987 \times 10^{-14}}{1 + \frac{\gamma_{H^+} 10^{-pH}}{\gamma_{NH_4^+}} H_{NH_3}^* W_i RT \times 0.987 \times 10^{-14}}$ (Eq. 2)

here, $[NH_4^+]$ is the molar concentration of NH$_4^+$ (mole m$^{-3}$). $\gamma$ is the activity coefficient, which is extracted from the

E-AIM IV model (Clegg et al., 1998a) to account for the non-ideality solution effect. $H_{NH_3}^*$ (atm$^{-1}$) represents the

molality-based equilibrium constant, which is T dependent and can be determined using equation (12) in Clegg et

al.(1998b). W$_i$ (μg m$^{-3}$) is the aerosol water content predicted by ISORROPIA-II. R (J/mole/K) is the universal gas

constant. T (K) is ambient temperature and 0.987$\times$10$^{-14}$ is the conversion multiplication factors from atm and μg to

SI units.

In Figure 6, ACR curve (The "S" shape curve, referred to as "S Curve" hereafter) is plotted against pH based on the

average T (10°C), AWC (100 μg m$^{-3}$), and $\frac{\gamma_{H^+}}{\gamma_{NH_4^+}}$ (2.4) during the haze period. Observation-based ACR as a function

of pH with varying T and AWC is also shown. Clearly, the observational ACR data points are relatively well

constrained by the theoretical equation, suggestive of reasonable judgement that ACR is controlled by T, AWC, pH,

and $\frac{\gamma_{H^+}}{\gamma_{NH_4^+}}$. Under the condition of average pH (4.6 ± 0.3) during the winter haze period, the "S curve" derives ACR

of 0.2, around half of the average measured ACR (0.4 ±0.1). Regional and long-range transport of aerosol pollutants

from northern China during the cold season (Xu et al., 2018) may have increased NH$_4^+$, yet NH$_3$ remains little

unaffected because of its limited transport distance (Asman et al., 1998). The transport effect cannot be predicted by

the theoretical equation and this should partly explain the divergence between the calculated and observed ACR.

Earlier works have also observed higher particle phase fraction than the henry's law constants predicted for water

soluble aerosol components (Arellanes et al., 2006;Hennigan et al., 2008;Shen et al., 2018). Another possible factor

contributing to the underestimation of ACR is the unaccounted effect from organic species, whose role in driving the

SNA formation is thought to be significant (Silvern et al., 2017). The organics have been found to account for 35%

of AWC in the southeast USA (Guo et al., 2015), thus ACR would be enhanced by including organic aerosol. To

quantitatively determine which parameter dominates the ACR, the impact on ACR from individual variable (i.e. T,





AWC, pH, and $\frac{\gamma_{H^+}}{\gamma_{NH_4^+}}$) during the haze period in winter is assessed (Figure 7). From a theoretical perspective, the

decrease of pH and T, and increase of AWC and $\frac{\gamma_{H^+}}{\gamma_{NH_4^+}}$ would raise ACR. For instance, in summertime, the lower

ACR (Figure 4) are mainly due to higher T that shift the equilibrium to the gas phase, thus higher $NH_3$ ($40 \pm 8$ µg m⁻

³) while lower $NH_4^+$ was observed. Likewise, in wintertime, the lower T facilitates the residence of $NH_4^+$ in the

particle phase than the gas phase ($NH_3$: $20 \pm 4$ µg m⁻³), resulting in higher ACR.

On the basis of "S curve" (Figure 7), each 0.1 unit change of ACR can be caused by approximate 5 °C, 75 µg m⁻³,

0.3, and 2 units change of T, AWC, pH, and $\frac{\gamma_{H^+}}{\gamma_{NH_4^+}}$, respectively. Actually, T, pH, and $\frac{\gamma_{H^+}}{\gamma_{NH_4^+}}$ are within a relatively

narrow range during the winter haze period (Table 1), suggesting the variation of these parameters shouldn't result

in the significant change of ACR. On the contrary, AWC fluctuates greatly during the study period (Table 1).

Therefore, AWC should be the key factor regulating ACR. It is well established that AWC is a function of RH and

atmospheric aerosol compositions (Pilinis et al., 1989;Wu et al., 2018;Nguyen et al., 2016;Hodas et al., 2014). AWC

has been known to promote secondary aerosol formation by providing aqueous medium for uptake of reactive gases,

gas to particle partitioning, and the subsequent chemical processing (McNeill, 2015;McNeill et al., 2012;Tan et al.,

2009;Xu et al., 2017b).

The winter haze pH in this study were ~3 units higher than that of the southeastern United States summer campaign

(Nah et al., 2018;Guo et al., 2015;Guo et al., 2017a;Xu et al., 2017a), but close to that of 3.7 in rural Europe (Guo et

al., 2018) and 4.2 in North China Plain (Liu et al., 2017), where $NH_3$-rich conditions are prevalent. AWC may act as

the major factor, because greater AWC dilute the [H⁺] and raise the pH. The AWC during the haze period ($82 \pm 105$

190    µg m⁻³) were much higher than those during the non-haze period ($32 \pm 41$ µg m⁻³).

**3.3 A possible self-amplifying feedback mechanism**

Given that AWC is a function of RH and SNA, a conceptual model of how AWC control ACR can be illustrated by

a self-amplifying feedback loop (Figure 8). Formation of SNA is initiated by gas-particle conversion of $NH_3$. Under

certain meteorological conditions such as high RH and shallow planetary boundary layer, SNA is subject to uptake

moisture and result in the increases of AWC. Based on the discussions above, the increase of AWC would further

raise ACR, leading to more efficient transformation of $NH_3$ as SNA.

Figure 9 shows the yearly average diurnal variation of ACR, AWC, SNA along with T and RH. Apparently, SNA

tracked well with ACR and AWC, especially over nighttime. The not well-correlated track between SNA and AWC

and ACR during the daytime (8:00-16:00) can be ascribed to the photochemical reactions that lead to SNA

formation. The good correlation between SNA and AWC and ACR demonstrated in figure 9 support the proposed

self-amplifying feedback loop in SNA formation.

**4 Conclusion**

Our results demonstrate that ACR, rather than $NH_3$ concentrations, plays a critical role in driving haze formation in

the agricultural $NH_3$ emitted regions. Based on the "S curve" calculation, we have unraveled that AWC is the major





factor controlling ACR. Upon analyzing the cross-correlations between AWC, ACR and SNA, we proposed a self-
amplifying feedback mechanism of SNA formation that associated with AWC and ACR. This positive feedback
cycle is likely to occur in other rural regions, where high agricultural $NH_3$ emissions are prevalent.

We have shown that high $NH_3$ concentrations may not necessarily lead to strong SNA formation, particularly in the
agriculture intensive areas, e.g. the North China Plain (NCP) and the extensive farming lands in Eastern China
where the high $NH_3$ levels are still unregulated and increasing (Meng et al., 2018;Warner et al., 2017). Although Liu
et al.(2019) have predicted that $PM_{2.5}$ can be slashed by 11-17% when 50% reduction in $NH_3$ from the agricultural
sector and 15% mitigation of $NO_x$ and $SO_2$ emissions was achieved, a recent study has demonstrated that only when
aerosol pH drops below 3.0, the $NH_3$ reduction would have expected mitigation effects (Guo et al., 2018). The
winter haze pH (4.6 ±0.3) in this study was mostly between 4-5. Our results thus imply that $NH_3$ only may not be an
effective solution to tackle air pollution in these regions.

***Data availability.***

The data presented in this paper are available upon request from the corresponding author (huangkan@fudan.edu.cn).

***Author contributions.***

JX and KH conceived the study. JX, JC, and KH performed data analysis and wrote the paper. All authors
contributed to the review of the manuscript.

***Competing interests.***

The authors declare that they have no conflict of interest.

***Acknowledgements***

The authors acknowledge support of the National Science Foundation of China (No. 91644105), the National Key
R&D Program of China (2018YFC0213105), and the Natural Science Foundation of Shanghai (19ZR1421100). Jian
Xu acknowledge project funded by China Postdoctoral Science Foundation (2019M651365).



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





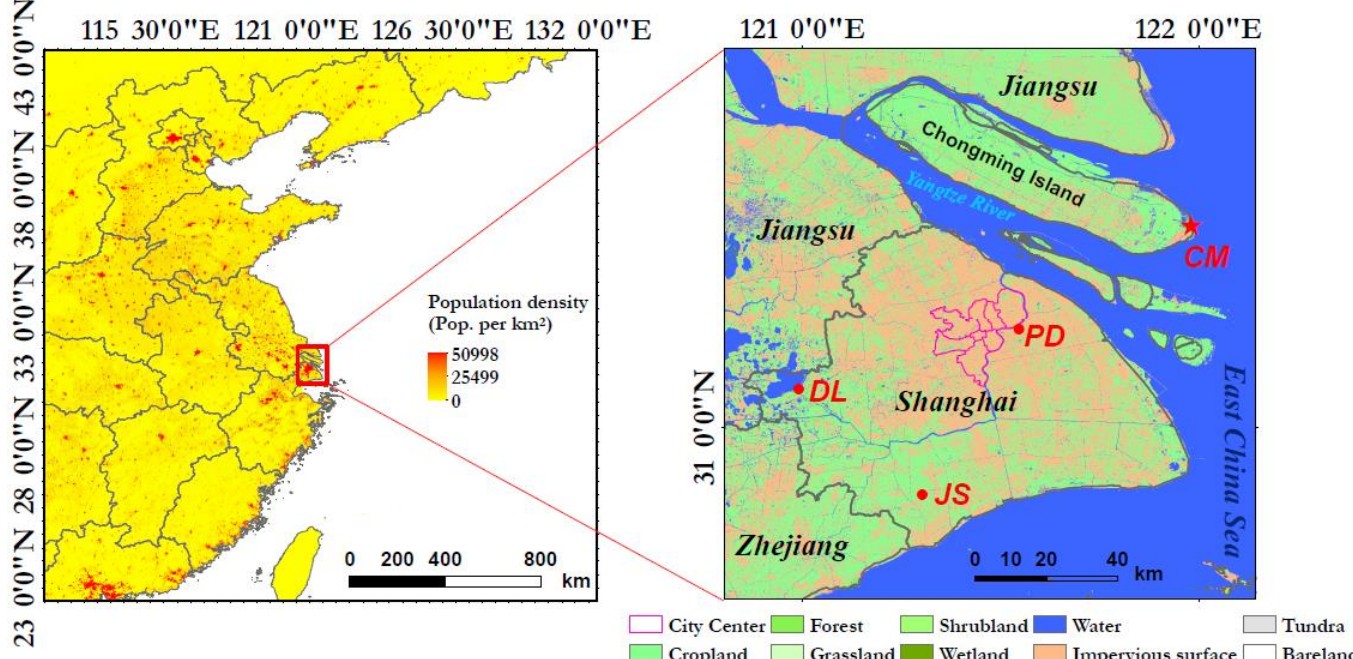

**Figure 1: Location of the sampling site. Population density is color-coded in the left panel. The right panel shows the land cover in Shanghai (adapted from Peng et al.(2018)). CM (Chongming) is the sampling site on Chongming island.**

**JS (Jinshan) represents the source emission from a dairy farm in rural Shanghai. DL (Dianshan Lake) represents a regional transport region in the Yangtze River Delta. PD (Pudong) represents the urban site.**





**Figure 2: Comparison of predicted and measured Cl⁻, NO₃⁻, NH₃, NH₄⁺. Orthogonal distance regression (ODR) fits with ±1σ are shown.**

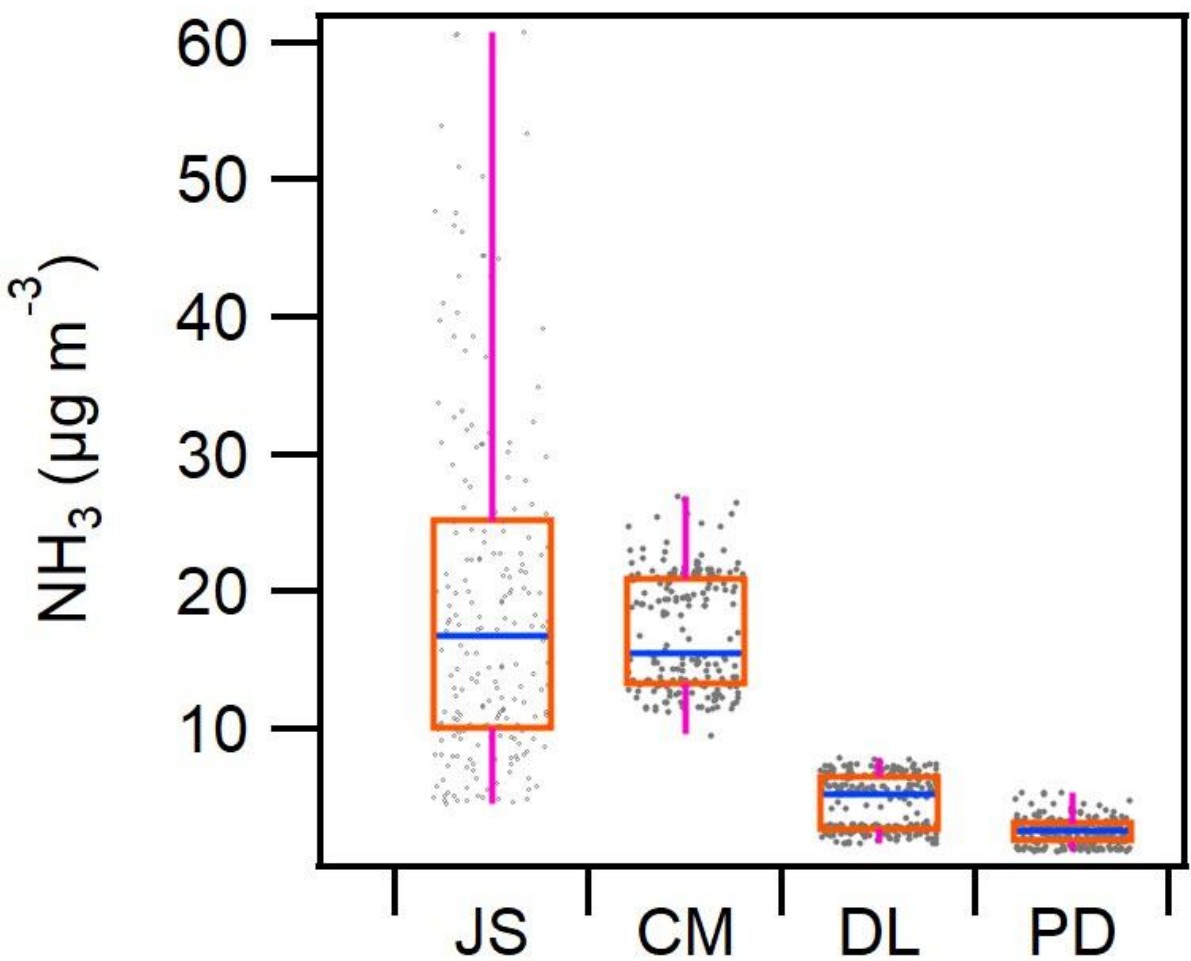

**Figure 3: NH₃ at different sampling site over the same period (From Jan 18 to 27 of 2018). The locations of all sites are shown in Figure 1. Scattered dots indicate raw data points.**

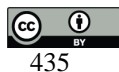

**Figure 4: Secondary inorganic aerosol mass concentration in PM$_{2.5}$ (SNA refers to sulfate, nitrate, and ammonium) as a function of NH$_3$. The sizes of the void circles are scaled to the ACR and colored by T. The SNA concentrations in black filled circles are binned and averaged according to the NH$_3$ mass concentration of each 10 μg m$^{-3}$. Error bars**
**represent one standard deviation (±1σ).**





**Figure 5: SNA mass concentration in PM₂.₅ as a function of ACR during the haze period. The circles are colored by the NH₃ mass concentration.**



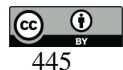

**Figure 6: ACR as a function of pH during the haze period. The sizes of the void circles are scaled to AWC and colored by T. The blue curve was calculated based on the average T (10 °C), AWC (100 μg m⁻³), and activity coefficients ratio of $\frac{\gamma_{H^+}}{\gamma_{NH_4^+}}$ respectively. The average $\frac{\gamma_{H^+}}{\gamma_{NH_4^+}}$ for the haze period is 2.4 ± 2.0 (±1σ).**



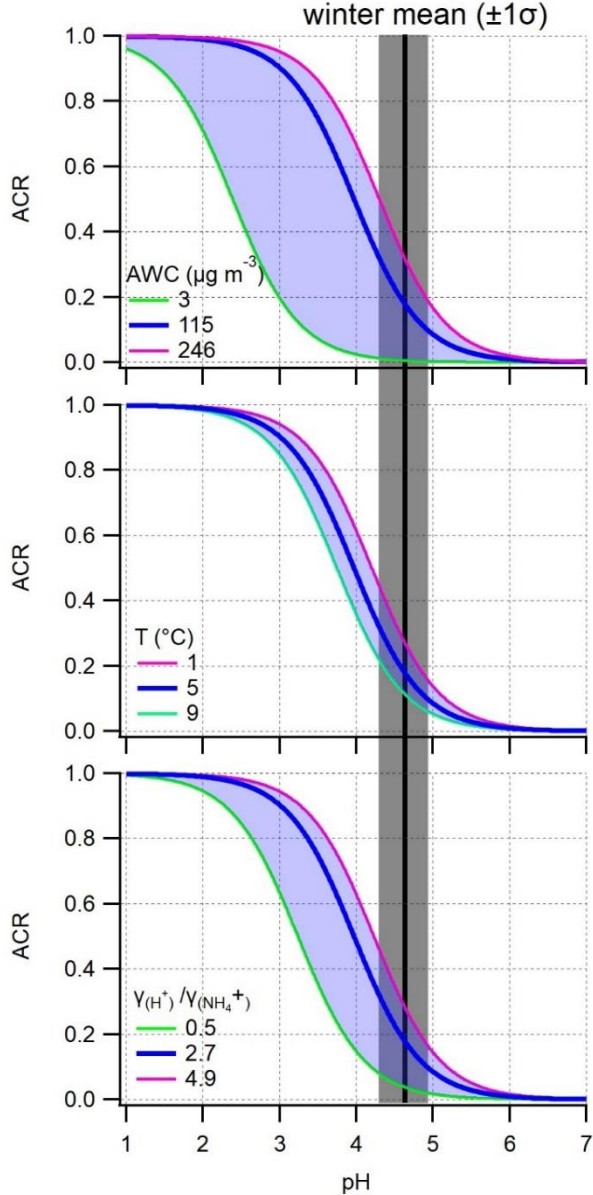

**Figure 7: ACR as a function of pH during the winter haze period. Other variables are held constant at the average value during the winter haze period, while varying only the AWC, T, activity coefficients ratio of** $\frac{\gamma_{H^+}}{\gamma_{NH_4^+}}$**, respectively. Shaded dark areas indicate the winter haze average pH together with one standard deviation (± 1σ). Shaded blue areas represent the curve corresponding to average ± 1σ, note that for AWC average - 1σ yield a negative value, thus the minimum mass concentration (3 μg m⁻³) was used.**



**Figure 8: Schematic of self-amplifying feedback loop for SNA formation.**

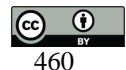



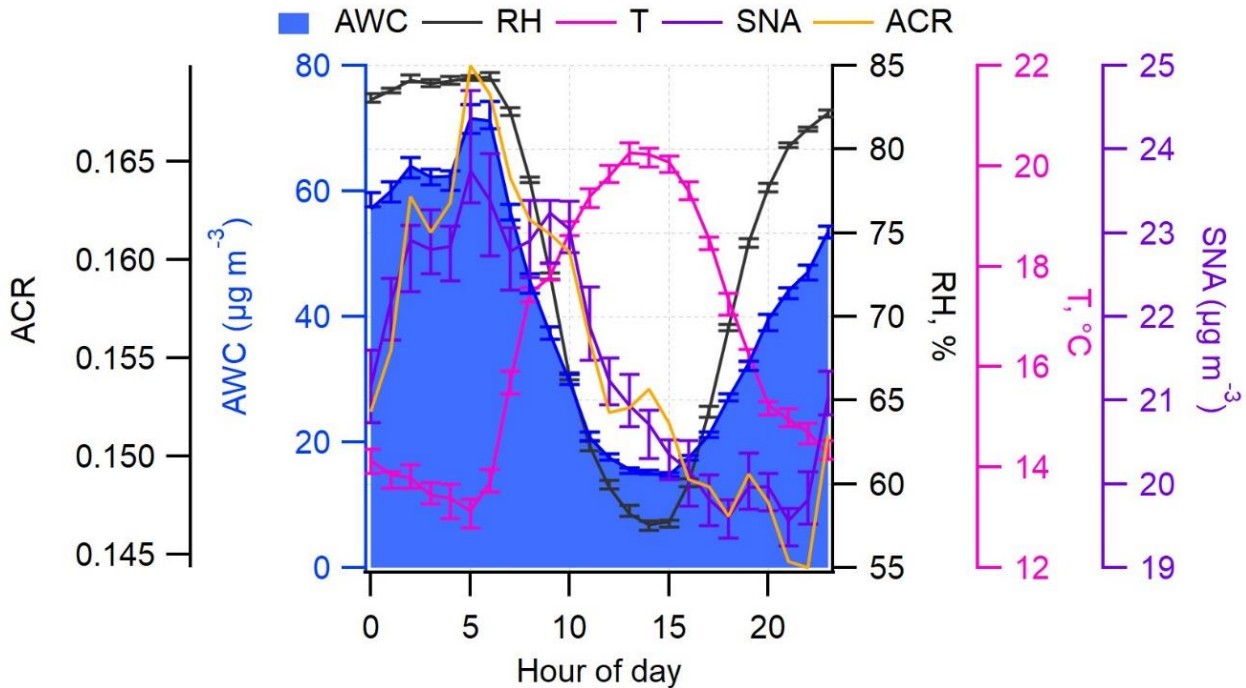

**Figure 9: Annual mean diurnal pattern of ACR, AWC, SNA, T, and RH.**



**Table 1: The summer and winter average (±1σ) ACR, pH, T, activity coefficients ratio of $\frac{\gamma_{H^+}}{\gamma_{NH_4^+}}$, and NH$_3$ (µg m$^{-3}$)**

**during the haze period.**

|  | ACR | AWC | pH | NH$_3$ | $\frac{\gamma_{H^+}}{\gamma_{NH_4^+}}$ | T |
|---|---|---|---|---|---|---|
| summer | 0.2 ±0.1 | 79 ±73 | 3.4 ±0.5 | 40 ±8 | 1.8 ±1.7 | 29 ±5 |
| winter | 0.4 ±0.1 | 115 ±131 | 4.6 ±0.3 | 20 ±4 | 2.7 ±2.2 | 5 ±4 |