# Peer review of "Importance of Gas-Particle Partitioning of Ammonia in Haze Formation in the Rural Agricultural Environment"

_Atmospheric Chemistry and Physics, 2019_

## Referee Comment (RC1) · Anonymous Referee #1 · 31 Jan 2020

This paper investigates the effect of NH3 concentration on NH3 partitioning to the particle phase and the influence of other factors, such as the concentration of a group of PM2.5 inorganic aerosol (sulfate, nitrate, ammonium; SNA) and ambient conditions in a region with very high NH3 plus NH4+ concentrations (ie, ammonia emissions). The title implies that the paper will investigate the overall influence of ammonia on PM2.5 haze, but this is never really done; eg, what fraction of the PM2.5 mass is ammonium, how does high ammonium affect the particle concentrations of Cl-, NO3-, what faction of the PM2.5 mass are these species? The analysis includes a thermodynamic model, with the focus of the paper solely on predictions of NH3/NH4+. Since the thermodynamic predictions depend on sulfate and nitrate, along with ammonium, (and possibly

chloride) these species should be included in the data presentation and discussion, but are largely ignored. For example, one should also present the partitioning (S curves) of nitrate, and possibly even chloride since it's concentrations are also fairly high (Fig 2) since these species are critical to the thermodynamic predictions and are highly hygroscopic and affect the aerosol liquid water. Overall the data is interesting, but the data analysis should be much more comprehensive and in depth.

Specific Comments

Was there even any mention of the sulfate concentration in this study?

The ACR is widely used in this paper and defined in the Abstract as the ammonia gas-particle conversion ratio, but that is ambiguous, explicitly define it in the Abstract. I strongly suggest the authors use a more common term, epsilon(NH4+), why make up new terminology?

How was the TEOM sample air dried? What was the temperature if dried? How would it affect mass concentrations of semivolatile species, such as NH4NO3?

Although PM2.5 mass and various gas phase species measurement methods are discussed, the data are really never considered in any important way, so why discuss the measurement method?

Figure 2. No explanation is given on why both gas/particle data are compared for ammonia but not for chloride or nitric acid. Also it would be insightful to plot ACR predicted vs measured, and plot and discuss the gas phase components of Cl- and NO3-, as was done for NH4+/HNO3. These comparisons, are also important to assess the model.

Since concentrations of the important other inorganic ions other than just NH4+ are not presented, the analysis in this paper is largely superficial. For example, roughly what is the form of the ammonium in the particle, is it ammonium nitrate, ammonium sulfate, ammonium chloride? Why not give a pie chart of the inorganic species concentrations,

as noted above regarding sulfate, no data on these other important species are given.

In Fig 4 the number of data points are not considered in the statistical results. Instead of plotting the error bar as the standard deviation plot it as the standard error, or better yet make bar-whisker plots instead. Does average = mean (mean is the more explicit term)? There is clearly a temp trend in this Fig, which should be explicitly discussed, ie, this plot simply shows that lower T more partitioning to the aerosol higher leading to high SNA and lower NH3 (if SNA is dominated by NA and not S).

Line 152. ISORROPIA is used for the predictions (ie, LWC, pH) and activity coefficients for the S curve are taken from E-AM. Is it reasonable to mix these two models? How do these activity coefficients compare to that predicted by ISORROPIA.

Line 135. The statement that NH3 concentration can be interpreted as the strength of the NH3 emissions is true only if only a small fraction of the total NH3 can be in the particle phase, which ACR shows is not true.

The table in the supplemental data is cut off and not all is legible.

Fig 6. The comparison between the S curve and the data is claimed to be good (relatively well constrained), but is it? Compare this result to other published identical plots and discuss why there appears to be more (or maybe less) discrepancy in this study. S curves of nitrate and possibly chloride should be included since the data exits and they can also be used to assess the thermodynamic model.

Lines 162-165. This explanation does not make sense. The model predicts the equilibrium state, which should exist at all time since the time scales to reach it are about 30 minutes. How will long range transport effect that. The authors need to investigate the thermodynamic model predictions better and come up with a better discussion.

Line 169. If the authors are going to accept the Silvern et al theory of a film impeding uptake of NH3, then they cannot accept the results of their thermodynamic analysis. Or they must assume that it affects only a fraction of the aerosol. This needs more

explanation.

Overall, the explanations for the discrepancy is largely just throwing out ideas and not assessing quantitatively the effect. For example, if the AWC is 35% higher how does that affect the S curve and data in Fig. 6. How will a different Henry's law affect the S curve in Fig 6 (ie which way is it shifted, to better or worse agreement with the data)?

Line 183 to 185. How does the affect of ALW on secondary aerosol formation (ie, I assume here from the references the authors are referring to SOA formation) affect the ACR?

Fig 9, ACR vertical axis is not colored the same way as the line on the plot (black and not yellow), whereas for the other plotted components there is consistency.

Section 3.3 Please explain the logic why similar diurnal trends in SNA, ACR and AWC at night supports the self amplifying feedback loop in SNA formation. Also how specifically does daytime photochemistry lead to a discrepancy if it is always assumed that the aerosol is in equilibrium?

SNA includes sulfate. How does sulfate play a role in this feedback mechanism?

The idea of feedback (or sometimes called co-condensation) leading to more uptake of NH3 by the added liquid water is not a new concept. It happens for any semi-volatile species that when partitioned to the particle phase significantly increases the water uptake. Thus, since sulfate is not semi-volatile and highly hygroscopic the species involved must generally have significantly higher concentrations then sulfate. Essentially here the effect is due to ammonium nitrate uptake, the same process discussed in Guo et al. (2017), yet here the focus is just on NH3/NH4+, the role of nitrate and possibly chloride in this process should also be included in the analysis. Note, that the molecular weight of NO3- is > NH4+ (thus most have focused on NO3- since one is generally concerned with effects on PM mass. Since the authors seem to think this is an important result from this work, they should discuss this process in much more detail and

cite other papers that have also investigated the process. See, for example Topping et al (2013).

Guo, H., J. Liu, K. D. Froyd, J. Roberts, P. R. Veres, P. L. Hayes, J. L. Jimenez, A. Nenes, and R. J. Weber (2017), Fine particle pH and gas-particle phase partitioning of inorganics in Pasadena, California, during the 2010 CalNex campaign, Atm. Chem. Phys., 17, 5703-5719.

Topping, D., P. Connolly, and G. McFiggans (2013), Cloud droplet number enhanced by co-condensation of organic vapours, Nature Geoscience, 6, 443-446.
* * *

---

## Referee Comment (RC2) · Anonymous Referee #3 · 19 Apr 2020

Secondary inorganic aerosol are major fractions of PM2.5 in China, especially during the hazy episode. Xu et al. investigate the role of ammonia gas-particle conversion ratio on secondary inorganic aerosol formation mechanisms in a rural site in China. They propose a self-amplifying feedback loop that link ammonia gas-particle conversion ratio with secondary inorganic aerosol. Overall, this paper makes a meaningful contribution to the haze formation mechanism in the rural agricultural areas in China. I favor its publication after the following issues are addressed.

1) Section 2.1 the authors have found the PM2.5 mass concentrations on Chongming site is higher than the urban (Pudong) site. Which fraction (sulfate, nitrate, ammonium

or organics) of PM2.5 mass is higher?

2) Section 3.1 the authors straightly go to NH3 levels in Chongming and its link to secondary inorganic aerosol. Since this article is about haze pollution, I would suggest them adding an overview section of the major PM2.5 species (NH4+, Na+, K+, Ca2+, Mg2+, Cl−, NO3−, and SO42−) to help the readers get a fully understanding about the typical air pollutants on the monitoring site.

3) Section 3.2 ISORROPIA II has been used to predict the pH and aerosol water, however, the activity coefficient extracted from E-AIM IV was adopted. The authors should provide comparison of activity coefficients predicted from the two models (Peng et al., 2019, EST).

4) Section 3.2 Lines 162-165, I suggest rephrasing the texts here.

5) Section 3.2 I suggest the authors compare their $\varepsilon$(NH4+) S curve results with previous reports in other sites using the same methodology (e.g., Figure4 in Nah et al., 2018, ACP).

---

## Author Response (AR1)

**Response to Reviewers' Comments**

We thank the reviewers for his/her careful reading of our manuscript and insightful comments and suggestions on greatly improving the quality of this manuscript. We address the referees' specific comments point-by-point below. The changes made to the revised manuscript were marked in orange.

**Referee #1**

This paper investigates the effect of NH3 concentration on NH3 partitioning to the particle phase and the influence of other factors, such as the concentration of a group of PM2.5 inorganic aerosol (sulfate, nitrate, ammonium; SNA) and ambient conditions in a region with very high NH3 plus NH4+ concentrations (ie, ammonia emissions). The title implies that the paper will investigate the overall influence of ammonia on PM2.5 haze, but this is never really done; eg, what fraction of the PM2.5 mass is ammonium, how does high ammonium affect the particle concentrations of Cl-, NO3-, what faction of the PM2.5 mass are these species?

Thanks for the comment. We have added a new section below in the Results and Discussion part to address the concern raised by the reviewer. The high ammonium concentration facilitates water uptake and enhance the aerosol water content. The elevated aerosol water serves as aqueous medium for uptake of reactive gases and promotes the gas-particle partitioning of semi-volatile species (e.g., HCl, HNO3, NH3 and certain organics), thus accelerating the mass growth of aerosol particles.

3.1 Overview of 1 year continuous measurements at Chongming

Figure 1 shows the time-series of hourly water-soluble $PM_{2.5}$ species during the study period. The mean concentration of $NO_3^-$, $SO_4^{2-}$, $NH_4^+$ and $Cl^-$ over the entire study period was 8.4 µg m$^{-3}$, 6.3 µg m$^{-3}$, 6.3 µg m$^{-3}$ and 1.2 µg m$^{-3}$. The haze period was defined as hourly averaged $PM_{2.5}$ mass loadings higher than 75 µg m$^{-3}$ and the rest was non-haze period. Table 1 gives the statistical summary of major aerosols during the haze and non-haze period. Clearly, the mass concentration of major $PM_{2.5}$ species ($NO_3^-$, $SO_4^{2-}$, $NH_4^+$ and $Cl^-$) increased during the haze period compared to those during the non-haze period. However, the concentration of NH3 showed no significant change during these two periods. The mean mass concentration of SNA (sulfate, nitrate, and ammonium) was 49.0 µg m$^{-3}$, contributing to about 50.0 % of total $PM_{2.5}$ mass.

Table 1. Statistical summary on mass concentrations of $PM_{2.5}$ species and NH3.

| Unit: µg m$^{-3}$ | $PM_{2.5}$ | $SO_4^{2-}$ | $NO_3^-$ | $Cl^-$ | $NH_4^+$ | NH3 |
|---|---|---|---|---|---|---|
| **non-haze** | 28.5 ± 16.9 | 5.6 ± 3.6 | 6.9 ± 6.6 | 1.1 ± 0.9 | 5.6 ± 3.3 | 32.2 ± 11.6 |
| **haze** | 98.3 ± 37.2 | 13.3 ± 7.7 | 23.1 ± 14.5 | 2.2 ± 1.9 | 13.2 ± 6.6 | 32.3 ± 13.5 |

[Figure]

Figure 1: Time series of $PM_{2.5}$ species during the study period.

The analysis includes a thermodynamic model, with the focus of the paper solely on predictions of NH3/NH4+. Since the thermodynamic predictions depend on sulfate and nitrate, along with ammonium, (and possibly chloride) these species should be included in the data presentation and discussion, but are largely ignored. For example, one should also present the partitioning (S curves) of nitrate, and possibly even chloride since it's concentrations are also fairly high (Fig 2) since these species are critical to the thermodynamic predictions and are highly hygroscopic and affect the aerosol liquid water. Overall the data is interesting, but the data analysis should be much more comprehensive and in depth.

Thanks for pointing out this. We agree with the reviewer that the partitioning (S curves) of nitrate, and chloride are critical to the thermodynamic predictions. However, it should be noted that the partitioning ratios (i.e., $\varepsilon(NO_3^-), \varepsilon(Cl^-)$) should be close to 50% in order to be useful for assessing the ISORROPIA II predictions(Guo et al., 2017). But this is not the case in this study especially for the winter haze period when gas phase HCl and gas phase $HNO_3$ only accounting for less than 4% of the total chloride and total nitrate. For example, the hourly HCl mass concentration was below the detection limit of the instrument during more than 70% of the winter season. And the detectable $HNO_3$ mass concentration in winter was around 0.4 µg m$^{-3}$ compared to about 11.8 µg m$^{-3}$ of nitrate in the particulate phase. Given that this paper focused on the haze pollution that mostly occurred in winter, we limited our discussion on the partitioning ratios of $\varepsilon(NH_4^+)$, which was close to 50% during the winter season as shown in figure 6.

In response, we plotted the following figure that shows the predicted vs measured $HNO_3$ and HCl, respectively. This figure has been added to the supplementary materials in the revision.

[Figure]

Figure R1: Comparison of predicted and measured $HNO_3$ and HCl in summer and in winter. Orthogonal distance regression (ODR) fits with ±1σ are shown.

**Specific Comments**

Was there even any mention of the sulfate concentration in this study?

Thanks for the comment. Sulfate concentration can be found in the new section 3.1. The mean concentration of $SO_4^{2-}$ over the entire study period was 6.3 µg m$^{-3}$. And the newly added table 1 gives the mean concentration of $SO_4^{2-}$ during the haze (13.3 ± 7.7 µg m$^{-3}$) and non-haze (5.6 ± 3.6 µg m$^{-3}$) period.

The ACR is widely used in this paper and defined in the Abstract as the ammonia gas-particle conversion ratio, but that is ambiguous, explicitly define it in the Abstract. I strongly suggest the authors use a more common term, epsilon(NH4+), why make up new terminology?

Thanks for pointing out this. We have changed ammonia gas-particle conversion ratio into $\varepsilon(NH_4^+)$ throughout the manuscript.

How was the TEOM sample air dried? What was the temperature if dried? How would it affect mass concentrations of semivolatile species, such as NH4NO3?

Thanks for the comment. TEOM sample air was dried through the filter dynamics measurement systems (FDMS). FDMS was equipped in TEOM 1405-F to account for both the volatile and non-volatile species of fine particles. According to the guidelines released by Thermo Scientific, *this is done by measuring the volatile portion of the sample independently from the total incoming sample, and using this fraction in calculating the PM mass concentration. FDMS dryer contains specially-designed Nafion tubing inlet on the main flow to minimize potential for particle loss. The dryer lowers the main flow relative humidity, and allows for mass transducer operation at 5 ℃ above the peak air monitoring station temperature. Purge Filter Conditioner contains a heat exchanger that maintains the temperature of the main air flow and particle filter at 4 ℃. An integrated humidity sensor that follows the SES dryer measures the main flow line humidity to determine the drying efficiency.*

Although PM2.5 mass and various gas phase species measurement methods are discussed, the data are really never considered in any important way, so why discuss the measurement method?

Thanks for pointing out this. The gas phase species measurement methods were deleted in the revision. Section 2.2 has been changed as below:

"……Meanwhile, ass loadings of PM$_{2.5}$ was determined by a Tapered Element Oscillating Microbalance coupled with Filter Dynamic Measurement System (TEOM 1405-F). ~~SO₂ mass concentrations were analyzed by Pulsed Fluorescence SO₂ Analyzer (Thermo Fisher Scientific, Model 43i). NO₂ mass concentrations were analyzed by Chemiluminescence NO-NO₂-NOₓ Analyzer (Thermo Fisher Scientific, Model 42i). O₃ mass concentrations were analyzed by UV Photometric Ozone Analyzer (Thermo Fisher Scientific, Model 49i). CO mass concentrations were analyzed by Gas Filter Correlation CO Analyzer (Thermo Fisher Scientific, Model 48i).~~ The QA/QC of these instruments were managed by professional staff in Shanghai Environmental Monitoring Center (SEMC) according to the Technical Guideline of Automatic Stations of Ambient Air Quality in Shanghai (HJ/T193-2005)."

Figure 2. No explanation is given on why both gas/particle data are compared for ammonia but not for chloride or nitric acid. Also it would be insightful to plot ACR predicted vs measured, and plot and discuss the gas phase components of Cl- and NO3-, as was done for NH4+/HNO3. These comparisons, are also important to assess the model.

Thanks for pointing out this. As responded above, the measured concentration of HNO$_3$ and HCl in the gas phase in winter was close to the detection limit of the instrument and much lower than the nitrate and chloride in the particle phase, respectively. Given that this paper focused on the haze pollution that mostly occurred in winter, we only show the gas/particle data comparison for ammonia but not for chloride or nitric acid in the original manuscript. In response, we plotted the following figure that shows the predicted vs measured HNO₃ and HCl, respectively. This figure has been added to the supplementary materials in the revision.

We agree with the reviewer that the partitioning (S curves) of nitrate, and chloride are critical to the thermodynamic predictions. And it should be noted that the partitioning ratios (i.e., $\varepsilon(NO_3^-)$, $\varepsilon(Cl^-)$ ) should be close to 50% in order to be useful for assessing the ISORROPIA II predictions. However, this is not the case in this study especially for the winter haze period when gas phase HCl and gas phase HNO₃ only accounting for less than 4% of the total chloride and total nitrate. For example, the hourly HCl mass concentration was below the detection limit of the instrument more than 70% of the winter season. And the detectable HNO₃ mass concentration in winter was around 0.4 µg m⁻³ compared to about 11.8 µg m⁻³ of nitrate in the particle phase. Given that this paper focused on the haze pollution that mostly occurred in winter, we limited our discussion on the partitioning ratios of $\varepsilon(NH_4^+)$, which was close to 50% during the winter season as shown in figure 6 in the manuscript.

[Figure]

Figure R1: Comparison of predicted and measured HNO₃ and HCl in summer and in winter. Orthogonal distance regression (ODR) fits with ±1σ are shown.

Since concentrations of the important other inorganic ions other than just NH4+ are not presented, the analysis in this paper is largely superficial. For example, roughly what is the form of the ammonium in the particle, is it ammonium nitrate, ammonium sulfate, ammonium chloride? Why not give a pie chart of the inorganic species concentrations, as noted above regarding sulfate, no data on these other important species are given.

Thanks for the comment. If we assume ammonium was preferably neutralized by sulfate than nitrate when NH₃ was in excess. Roughly 47% of ammonium was expected to be in the form of ammonium sulfate during the winter haze period. And ammonium nitrate dominates the remaining part of ammonium. We have added a new section below in the Results and Discussion that shows inorganic species concentrations.

3.1 Overview of 1 year continuous measurements on Chongming
Figure 1 shows the time-series of hourly water-soluble PM$_{2.5}$ species during the study period. The mean concentration of NO$_3^-$, SO$_4^{2-}$, NH$_4^+$ and Cl$^-$ over the entire study period was 8.4 μg m$^{-3}$, 6.3 μg m$^{-3}$, 6.3 μg m$^{-3}$ and 1.2 μg m$^{-3}$. The haze period was defined as hourly averaged PM$_{2.5}$ mass loadings higher than 75 μg m$^{-3}$ and the rest was non-haze period. Table 1 gives the statistical summary of major aerosols during the haze and non-haze period. Clearly, the mass concentration of major PM$_{2.5}$ species (NO$_3^-$, SO$_4^{2-}$, NH$_4^+$ and Cl$^-$) increased during the haze period compared to those during the non-haze period. However, the concentration of NH$_3$ showed no significant change during these two periods. The mean mass concentration of SNA (sulfate, nitrate, and ammonium) was 49.0 μg m$^{-3}$, contributing to 50.0 % of total PM$_{2.5}$ mass.

Table 1. Statistical summary on mass concentrations of PM$_{2.5}$ species and NH$_3$.

| Unit: μg m$^{-3}$ | PM$_{2.5}$ | SO$_4^{2-}$ | NO$_3^-$ | Cl$^-$ | NH$_4^+$ | NH$_3$ |
|---|---|---|---|---|---|---|
| non-haze | 28.5 ± 16.9 | 5.6 ± 3.6 | 6.9 ± 6.6 | 1.1 ± 0.9 | 5.6 ± 3.3 | 32.2 ± 11.6 |
| haze | 98.3 ± 37.2 | 13.3 ± 7.7 | 23.1 ± 14.5 | 2.2 ± 1.9 | 13.2 ± 6.6 | 32.3 ± 13.5 |

[Figure]

Figure 1: Time series of PM$_{2.5}$ species during the study period.

In Fig 4 the number of data points are not considered in the statistical results. Instead of plotting the error bar as the standard deviation plot it as the standard error, or better yet make bar-whisker plots instead. Does average = mean (mean is the more explicit term)? There is clearly a temp trend in this Fig, which should be explicitly discussed, ie, this plot simply shows that lower T more partitioning to the aerosol higher leading to high SNA and lower NH3 (if SNA is dominated by NA and not S).

Thanks for the comment. We have added the number of data points in Figure 4. The bar-whisker plot has been inserted into the new figure 4 as below in the revision. Yes, here "average" means "mean" and the term "average" is replaced by "mean" throughout the manuscript. We did a statistical analysis of SNA concentration versus temperature (T). T has been divided into <0, 0~10, 10~20, 20~30, and >30 °C, the resulting mean SNA concentration was 21.1, 24.8, 21.3, 17.9 and 20.0 μg m$^{-3}$, respectively. Hence, there were no significant differences of SNA under different bins of temperature. We agree with the reviewer that temperature plays an important role in the SNA formation. However, as we pointed out throughout the MS, other factors including pH, aerosol water content and activity coefficients ratio of $\frac{\gamma_{H^+}}{\gamma_{NH_4^+}}$ also affect the SNA formation.

[Figure]

Line 152. ISORROPIA is used for the predictions (ie, LWC, pH) and activity coefficients for the S curve are taken from E-AM. Is it reasonable to mix these two models? How do these activity coefficients compare to that predicted by ISORROPIA.

Thanks for the comment. We have re-calculated and re-plotted the S curve using activity coefficients taken from ISORROPIA II with the help of our co-author Dr. Hongyu Guo. For example, figure 6 has been replotted with $\frac{\gamma_{H^+}}{\gamma_{NH_4^+}}$ predicted by ISORROPIA II in the revision. Note the S curve only shifted to the right after the input of $\frac{\gamma_{H^+}}{\gamma_{NH_4^+}}$ changed from 2.4 ± 2.0 to 4.0 ± 2.6.

[Figure]

Figure 6: $\varepsilon(NH_4^+)$ as a function of pH during the haze period. The sizes of the void circles are scaled to AWC and colored by T. The blue curve was calculated based on the average T (10 ℃), AWC (100 μg m⁻³), and activity coefficients ratio of $\frac{\gamma_{H^+}}{\gamma_{NH_4^+}}$ respectively. The average $\frac{\gamma_{H^+}}{\gamma_{NH_4^+}}$ for the haze period is 4.0 ± 2.6 (±1σ).

Line 135. The statement that NH3 concentration can be interpreted as the strength of the NH3 emissions is true only if only a small fraction of the total NH3 can be in the particle phase, which ACR shows is not true.
Thanks for the comment. It should be noted that $NH_3$ means gas phase ammonia in the manuscript.

The table in the supplemental data is cut off and not all is legible.
Thanks for pointing out this. We have re-sized the columns of Table S1 in the revised version.

Fig 6. The comparison between the S curve and the data is claimed to be good (relatively well constrained), but is it? Compare this result to other published identical plots and discuss why there appears to be more (or maybe less) discrepancy in this study. S curves of nitrate and possibly chloride should be included since the data exits and they can also be used to assess the thermodynamic model.
Thanks for the suggestion. The S curve in Fig. 6 represents the relationship between $\varepsilon(NH_4^+)$ and pH based on the mean condition of the winter haze period, i.e. T of 10 ℃ and AWC of 100 µg m-3. As the observational data points in Fig. 6 covered data during the whole study period, it is reasonable that some data points didn't distribute along the curve. Using the same methodology as figure 4 in Nah et al., 2018, we picked a small range of AWC (80 to 120 µg m$^{-3}$) and T (8 to 12 ℃) to be close to the average AWC (100 µg m$^{-3}$) and T (10 ℃) during the haze period. We calculated the S curve of $\varepsilon(NH_4^+)$ with the average AWC and T and plot the selected dataset in Figure R2 as below. Clearly, for the data points selected, there is roughly the same amount of spread compared to Nah et al., 2018.

We agree with the reviewer that the partitioning (S curves) of nitrate, and chloride are critical to the thermodynamic predictions. However, note that the partitioning ratios (i.e., $\varepsilon(NO_3^-), \varepsilon(Cl^-)$ ) should be close to 50% in order to be useful for assessing the ISORROPIA II predictions(Guo et al., 2017). But this is not the case in this study especially for the winter haze period when gas phase HCl and gas phase $HNO_3$ only accounting for less than 4% of the total chloride and total nitrate. For example, the hourly HCl mass concentration was below the detection limit of the instrument more than 70% of the winter season. And the detectable $HNO_3$ mass concentration in winter was around 0.4 µg m$^{-3}$ compared to about 11.8 µg m$^{-3}$ of nitrate in the particle phase. Given that this paper focused on the haze pollution that mostly occurred in winter, we limited our discussion on the partitioning ratios of $\varepsilon(NH_4^+)$, which was close to 50% during the winter season as shown in figure 6.

[Figure]

Figure R2: Analytically calculated S curves of $\varepsilon(NH_4^+)$ (Black curve) and measured $\varepsilon(NH_4^+)$ (void green circles) as a function of pH. A small range of AWC (80 to 120 µg m$^{-3}$) and T (8 to 12 ℃) to be close to the average AWC (100 µg m$^{-3}$) and T (10 ℃) was selected during the haze period. The S curve calculated based on the average T (10 ℃), AWC (100 µg m$^{-3}$), and activity coefficients ratio of $\frac{\gamma_{H^+}}{\gamma_{NH_4^+}}$

(6.0) respectively. Note the average $\frac{\gamma_{H^+}}{\gamma_{NH_4^+}}$ for the selected ambient data we used to calculate the S curve is 6.0 not 4.0.

Thanks for pointing out this. We agree with the reviewer that long range transport would not affect the equilibrium. After re-calculating the possible effect from organic mass as shown in Figure R3 below, we have deleted these texts.

Thanks for the comment. Silvern et al., 2017 suggested that inorganics particles are coated by organic film, impeding the uptake of ammonia. The presence of organics was expected to slow down the achievements of inorganic thermodynamic equilibrium. We agree with the reviewer that the organic coating affects only a fraction of the inorganic aerosol and the thermodynamic equilibrium were assumed to achieve within one hour even organic films present.

Overall, the explanations for the discrepancy is largely just throwing out ideas and not assessing quantitatively the effect. For example, if the AWC is 35% higher how does that affect the S curve and data in Fig. 6. How will a different Henry's law affect the S curve in Fig 6 (ie which way is it shifted, to better or worse agreement with the data)?
Thanks for the comment. The discrepancy has been quantitatively analyzed as below in the revision. Since the mass concentration of organic aerosol was not available in this study, We did a sensitivity analysis via increasing the AWC by 10, 20 to 90 µg m$^{-3}$ as shown in Figure R3. The pH was not re-calculated using the new AWC because the co-existed organic aerosol altered pH in a complex way (Battaglia Jr et al., 2019;Wang et al., 2018;Pye et al., 2020). For example, some organic acids increase aerosol acidity thus decrease pH, whereas organic basics (e.g., amines) raise aerosol pH. We found that the best agreement between the predicted and measured $\varepsilon(NH_4^+)$ was achieved when we increase the AWC by roughly 90 µg m$^{-3}$, suggesting a nearly 48% of AWC contributed by the organics. This result falls in the range from a recent report in North China that organics contribute to 30 % ± 22% of AWC (Jin et al., 2020), and slightly higher than those southeastern United States sites that organic aerosol-related water accounting for about 29 to 39% of total water (Guo et al., 2015) and those in the eastern Mediterranean that about 27.5% of total aerosol water resulted from organics (Bougiatioti et al., 2016).
A higher Henry's law constant shift the S curve to the right and a lower Henry's law constant shift the S curve to the left.

[Figure]

Figure R3: Comparison of predicted and measured $\varepsilon(NH_4^+)$. Note the predicted $\varepsilon(NH_4^+)$ was analytically calculated using the equation 2 with input (i.e., pH, AWC, $\frac{\gamma_{H^+}}{\gamma_{NH_4^+}}$) taken from ISORROPIA II prediction and the AWC has been increased by 10, 20 to 90 µg m$^{-3}$ while other inputs fixed. Orthogonal distance regression (ODR) fits line (red) and y=x line (dashed orange) was shown for the clarity of the figure.

Line 183 to 185. How does the affect of ALW on secondary aerosol formation (ie, I assume here from the references the authors are referring to SOA formation) affect the ACR?

Thanks for the comment. The authors want to mention the AWC has also been known to enhance SOA formation. However, the enhanced SOA formation resulting from AWC may not affect the ACR or possible affecting the ACR in the same way as those semi-volatile inorganics.

The text has been revised as "AWC has also been known to promote secondary organic aerosol formation by providing aqueous medium for uptake of reactive gases, gas to particle partitioning, and the subsequent chemical processing (McNeill, 2015;McNeill et al., 2012;Tan et al., 2009;Xu et al., 2017)."

Fig 9, ACR vertical axis is not colored the same way as the line on the plot (black and not yellow), whereas for the other plotted components there is consistency.

Thanks for pointing out this. The ACR vertical axis has been changed to yellow color as below in the revision.

[Figure]

Section 3.3 Please explain the logic why similar diurnal trends in SNA, ACR and AWC at night supports the self amplifying feedback loop in SNA formation. Also how specifically does daytime photochemistry lead to a discrepancy if it is always assumed that the aerosol is in equilibrium?

Thanks for the comment. We understand that if the self-amplifying feedback mechanism dominate the SNA formation, then the diurnal trends of SNA, ACR and AWC would track each other. In Figure 9, we saw similar diurnal trends in SNA, ACR and AWC at night. So the self-amplifying feedback mechanism can be verified during the nighttime. But the daytime SNA concentration did not show similar variation as ACR and AWC, so we expect other mechanism also in play. Since the trend deviation of SNA concentration happened during the mid-afternoon when strong photochemical activity occurs. Therefore, we assume both the self-amplifying feedback mechanism and photochemical production was in play during the daytime while self-amplifying feedback mechanism dominate the SNA formation at night.

SNA includes sulfate. How does sulfate play a role in this feedback mechanism?

Thanks for the comment. SNA includes sulfate, and after the initial formation of SNA triggered by the gas-particle conversion of $NH_3$, the sulfate, together with nitrate and ammonium, promotes water uptake and resulted in the increase of aerosol water content. Then the increase of aerosol water content further raise the $\varepsilon(NH_4^+)$ and more SNA formed. So the role of sulfate in the feedback mechanism is to facilitate water uptake.

The idea of feedback (or sometimes called co-condensation) leading to more uptake of NH3 by the added liquid water is not a new concept. It happens for any semi-volatile species that when partitioned to the particle phase significantly increases the water uptake. Thus, since sulfate is not semi-volatile and highly hygroscopic the species involved must generally have significantly higher concentrations then sulfate. Essentially here the effect is due to ammonium nitrate uptake, the same process discussed in Guo et al. (2017), yet here the focus is just on NH3/NH4+, the role of nitrate and possibly chloride in this process should also be included in the analysis. Note, that the molecular weight of NO3is > NH4+ (thus most have focused on NO3since one is generally concerned with effects on PM mass. Since the authors seem to think this is an important result from this work, they should discuss this process in much more detail and cite other papers that have also investigated the process. See, for example Topping et al (2013).

Guo, H., J. Liu, K. D. Froyd, J. Roberts, P. R. Veres, P. L. Hayes, J. L. Jimenez, A. Nenes, and R. J. Weber (2017), Fine particle pH and gas-particle phase partitioning of inorganics in Pasadena, California, during the 2010 CalNex campaign, Atm. Chem. Phys., 17, 5703-5719.

Topping, D., P. Connolly, and G. McFiggans (2013), Cloud droplet number enhanced by co-condensation of organic vapours, Nature Geoscience, 6, 443-446.

Thanks for the comment. We agree with the points raised by this reviewer that "since sulfate is not semi-volatile and highly hygroscopic the species involved must generally have significantly higher concentrations then sulfate". Nitrate is more hygroscopic than sulfate and the mass concentration of nitrate was indeed significantly higher than sulfate in this study (Table 1). And the work by Topping (Topping et al., 2013) has been cited in the revision below.

Given that AWC is a function of RH and SNA, a conceptual model of how AWC control ACR can be illustrated by a self-amplifying feedback loop (Figure 8). Formation of SNA is initiated by gas-particle conversion of NH$_3$. Under certain meteorological conditions such as high RH and shallow planetary boundary layer, SNA is subject to uptake moisture and result in the increases of AWC. The enhanced aerosol water dilutes the vapor pressure of semi-volatile species (i.e., nitrate, ammonium and chloride) above the particle and driving semi-volatile species continue to condense (Topping et al., 2013). Based on the discussions above, the increase of AWC would further raise ACR, leading to more efficient transformation of NH$_3$ as SNA.

**Referee #3**

Secondary inorganic aerosol are major fractions of PM2.5 in China, especially during the hazy episode. Xu et al. investigate the role of ammonia gas-particle conversion ratio on secondary inorganic aerosol formation mechanisms in a rural site in China. They propose a self-amplifying feedback loop that link ammonia gas-particle conversion ratio with secondary inorganic aerosol. Overall, this paper makes a meaningful contribution to the haze formation mechanism in the rural agricultural areas in China. I favor its publication after the following issues are addressed.

Thanks for the positive comments.

1) Section 2.1 the authors have found the PM2.5 mass concentrations on Chongming site is higher than the urban (Pudong) site. Which fraction (sulfate, nitrate, ammonium or organics) of $PM_{2.5}$ mass is higher?

Thanks for the comment. In this study, the organic mass was not measured on Chongming site. The mass concentration of ammonium on Chongming ($6.2\pm4.5$ µg m$^{-3}$) is higher than that on Pudong ($5.0\pm5.5$ µg m$^{-3}$). The sulfate ($6.5\pm4.9$ µg m$^{-3}$) and nitrate ($7.6\pm9.1$ µg m$^{-3}$) concentration on Chongming is slightly lower than the level of sulfate ($6.8\pm5.7$ µg m$^{-3}$) and nitrate ($8.9\pm11.3$ µg m$^{-3}$) on Pudong.

2) Section 3.1 the authors straightly go to $NH_3$ levels in Chongming and its link to secondary inorganic aerosol. Since this article is about haze pollution, I would suggest them adding an overview section of the major PM2.5 species (NH4+, Na+, K+, Ca2+, Mg2+, Cl-, NO3-, and SO42-) to help the readers get a fully understanding about the typical air pollutants on the monitoring site.

Thanks for the comment. We have added a new section below in the Results and Discussion part to address the questions raised by the reviewer.

3.1 Overview of 1 year continuous measurements on Chongming

Figure 1 shows the time-series of hourly water-soluble $PM_{2.5}$ species during the study period. The mean concentration of $NO_3^-$, $SO_4^{2-}$, $NH_4^+$ and $Cl^-$ over the entire study period was 8.4 µg m$^{-3}$, 6.3 µg m$^{-3}$, 6.3 µg m$^{-3}$ and 1.2 µg m$^{-3}$. The haze period was defined as hourly averaged $PM_{2.5}$ mass loadings higher than 75 µg m$^{-3}$ and the rest was non-haze period. Table 1 gives the statistical summary of major aerosols during the haze and non-haze period. Clearly, the mass concentration of major $PM_{2.5}$ species ($NO_3^-$, $SO_4^{2-}$, $NH_4^+$ and $Cl^-$) increased during the haze period compared to those during the non-haze period. However, the concentration of $NH_3$ showed no significant change during these two periods. The mean mass concentration of SNA (sulfate, nitrate, and ammonium) was 49.0 µg m$^{-3}$, contributing to 50.0 % of total $PM_{2.5}$ mass.

Table 1. Statistical summary on mass concentrations of $PM_{2.5}$ species and $NH_3$.

| Unit: µg m$^{-3}$ | PM$_{2.5}$ | SO$_4^{2-}$ | NO$_3^-$ | Cl$^-$ | NH$_4^+$ | NH$_3$ |
|---|---|---|---|---|---|---|
| non-haze | 28.5 ± 16.9 | 5.6 ± 3.6 | 6.9 ± 6.6 | 1.1 ± 0.9 | 5.6 ± 3.3 | 32.2 ± 11.6 |
| haze | 98.3 ± 37.2 | 13.3 ± 7.7 | 23.1 ± 14.5 | 2.2 ± 1.9 | 13.2 ± 6.6 | 32.3 ± 13.5 |

[Figure]

Figure 1: Time series of PM$_{2.5}$ species during the study period.

3) Section 3.2 ISORROPIA II has been used to predict the pH and aerosol water, however, the activity coefficient extracted from E-AIM IV was adopted. The authors should provide comparison of activity coefficients predicted from the two models (Peng et al., 2019, EST).

Thanks for the comment. We have re-calculated and re-plotted the S curve using activity coefficients taken from ISORROPIA II with the help of our co-author Dr. Hongyu Guo. For example, the following figure 6 has been replotted with $\frac{\gamma_{H^+}}{\gamma_{NH_4^+}}$ predicted by ISORROPIA II in the revision. Note the S curve only shifted to the right after the input of $\frac{\gamma_{H^+}}{\gamma_{NH_4^+}}$ changed from 2.4 ± 2.0 to 4.0 ± 2.6.

[Figure]

Figure 6: $\varepsilon(NH_4^+)$ as a function of pH during the haze period. The sizes of the void circles are scaled to AWC and colored by T. The blue curve was calculated based on the average T (10 °C), AWC (100 µg m$^{-3}$), and activity coefficients ratio of $\frac{\gamma_{H^+}}{\gamma_{NH_4^+}}$ respectively. The average $\frac{\gamma_{H^+}}{\gamma_{NH_4^+}}$ for the haze period is 4.0 ± 2.6 (±1σ).

4) Section 3.2 Lines 162-165, I suggest rephrasing the texts here.

Thanks for the comment. We noticed that the long range transport would not affect the equilibrium. After re-calculated the possible effect from organic mass, we have deleted these texts.

5) Section 3.2 I suggest the authors compare their ε(NH4+) S curve results with previous reports in other sites using the same methodology (e.g., Figure4 in Nah et al., 2018, ACP).

Thanks for the comment. A narrow range of aerosol water content (1 to 4 µg m$^{-3}$) and temperature (15 to 25 °C) in their ambient dataset was selected to be close to the input (AWC=2.5 µg m$^{-3}$ and T = 20 °C) of analytically calculated S curves. However, in this study, the average AWC and T during the winter haze period was 100 µg m$^{-3}$ and 10 °C, respectively. So a direct comparison to Nah et al., should be discouraged. Using the same methodology, we picked a small range of AWC (80 to 120 µg m⁻³) and T (8 to 12 °C) to be close to the average AWC (100 µg m⁻³) and T (10 °C) during the haze period. We calculated the S curve of $\varepsilon(NH_4^+)$ with the average AWC and T and plot the data as below. Clearly, for the data points selected, there is roughly the same amount of spread compared to Nah et al., 2018.

[Figure]

Figure R2: Analytically calculated S curves of $\varepsilon(NH_4^+)$ (Black curve) and measured $\varepsilon(NH_4^+)$ (void green circles) as a function of pH. A small range of AWC (80 to 120 µg m⁻³) and T (8 to 12 °C) to be close to the average AWC (100 µg m⁻³) and T (10 °C) was selected during the haze period. The S curve was calculated based on the average T (10 °C), AWC (100 µg m⁻³), and activity coefficients ratio of $\frac{\gamma_{H^+}}{\gamma_{NH_4^+}}$ (6.0) respectively. Note the average $\frac{\gamma_{H^+}}{\gamma_{NH_4^+}}$ for the selected ambient data we used to calculate the S curve is 6.0 not 4.0.

**Importance of Gas-Particle Partitioning of Ammonia in Haze Formation in the Rural Agricultural Environment**

Jian Xu[1], Jia Chen[1], Na Zhao[1], Guochen Wang[1], Guangyuan Yu[1], Hao Li[1], Juntao Huo[3], Yanfen Lin[3], Qingyan Fu[3], Hongyu Guo[4], Congrui Deng[1], Shan-Hu Lee[5], Jianmin Chen[1], Kan Huang[1,2]*

[1]Shanghai Key Laboratory of Atmospheric Particle Pollution and Prevention (LAP3), Department of Environmental Science and Engineering, Fudan University, Shanghai 200433, China
[2]Institute of Eco-Chongming (IEC), No.20 Cuiniao Road, Chen Jiazhen, Shanghai, China, 202162

[revised manuscript text omitted]

S2).

**3 Results and Discussion**

**3.1 Overview of 1 year continuous measurements at Chongming**

Figure 2 shows the time-series of hourly water-soluble $PM_{2.5}$ species during the study period. The mean concentration of $NO_3^-$, $SO_4^{2-}$, $NH_4^+$ and $Cl^-$ over the entire study period was 8.4 μg $m^{-3}$, 6.3 μg $m^{-3}$, 6.3 μg $m^{-3}$ and

1.2 μg m$^{-3}$. The haze period was defined as hourly mean PM$_{2.5}$ mass loadings higher than 75 μg m$^{-3}$ and the rest was non-haze periods. Table 1 gives the statistical summary of major aerosols during the haze and non-haze period.

Clearly, the mass concentration of major PM$_{2.5}$ species (NO$_3^-$, SO$_4^{2-}$, NH$_4^+$ and Cl$^-$) increased during the haze period compared to those during the non-haze period. However, the concentration of NH$_3$ showed no significant change during these two periods. The mean mass concentration of SNA (sulfate, nitrate, and ammonium) was 49.0

μg m$^{-3}$, contributing to about 50.0 % of total PM$_{2.5}$ mass.

**3.2 NH$_3$ levels and its link to secondary inorganic aerosol**

Figure 3 shows that the mean concentration of NH$_3$ at Chongming (CM: 17.0 ± 4.2 μg m$^{-3}$) was more than three times higher than an urban site in Shanghai (PD: 2.5 ± 0.9 μg m$^{-3}$) and a representative regional transport region (DL: 4.6 ± 2.0 μg m$^{-3}$) in the Yangtze River Delta. The level of NH$_3$ at Chongming was even close to that observed inside a typical dairy farm (JS: 19.4 ± 12.6 μg m$^{-3}$), which was dominated by livestock emissions. Thus, it is interesting to investigate how the formation of secondary inorganic aerosols is impacted by this abnormally high level of NH$_3$.

Figure 4 indicates the response of SNA (sulfate, nitrate, and ammonium) mass concentrations to NH$_3$ is nonlinear.

Higher NH$_3$ sometimes correspond to even lower SNA mass concentrations. Statistically, the mean SNA

concentration in each bin of NH$_3$ doesn't show significant difference. This is at odds with the traditional view that higher concentrations of precursors usually result in elevated inorganic aerosols (Nowak et al., 2010). Although the abundance of SNA is related to the alkaline gaseous precursor (e.g., NH$_3$), the ambient condition (e.g., RH and T), and acid precursors (i.e., SO$_2$ and NO$_x$) whether favor the conversion of precursors into particles or not is equally important, if not higher. For example, the urban areas show higher SNA levels than the rural region while lower

NH$_3$ mixing ratio was observed (Wu et al., 2016;Wang et al., 2015b). Previous field measurements suggest that rural

NH$_4^+$ levels were more sensitive to acidic gases than to the NH$_3$ availability (Shen et al., 2011;Robarge et al., 2002).

Therefore, the level of NH$_3$ concentration is not the determining factor for the formation of secondary inorganic aerosols.

**3.3 The role of $\varepsilon(NH_4^+)$**

In this regard, we further investigate the relationship between the gas-particle partitioning of ammonia ($\varepsilon(NH_4^+)$, defined as the molar ratio between particle phase ammonia (NH$_4^+$) and total ammonia (NH$_x$ = NH$_3$+NH$_4^+$)) and SNA

during the haze period. The haze period is defined as hourly mean PM$_{2.5}$ mass loadings higher than 75 μg m$^{-3}$.

As shown in Figure 5, it is obvious that SNA in PM$_{2.5}$ is almost linearly correlated with $\varepsilon(NH_4^+)$. Higher $\varepsilon(NH_4^+)$

results in higher SNA concentrations. In addition, under the same $\varepsilon(NH_4^+)$ conditions, higher NH$_3$ promotes stronger formation of SNA. Thus, NH$_3$ and $\varepsilon(NH_4^+)$ collectively determine the haze formation potential. The level of NH$_3$ can be regarded as a proxy of NH$_3$ emission intensity, which is source dependent. As for $\varepsilon(NH_4^+)$, it represents the relative abundance of gaseous NH$_3$ and particulate ammonium. The shift between the two phases is controlled by various factors such as the ambient environmental conditions. Previous study shows that elevated RH

and acidic gas levels favor the shift of NH$_3$ towards the particulate phase at an urban site, thereby a lower

[NH$_3$]:[NH$_4^+$] ratio was observed (Wei et al., 2015). In this study, it is also observed that higher $\varepsilon(NH_4^+)$ values coincide with heightened RH, SO$_2$, and NO$_x$.

Based on the above results, elucidation of the driving factors determining $\varepsilon(NH_4^+)$ is of great importance to explore the formation mechanism of haze. Theoretically, $\varepsilon(NH_4^+)$ is determined by NH$_3$, NH$_4^+$, and the equilibrium between NH$_3$ and NH$_4^+$. Assuming NH$_3$ and NH$_4^+$ are in thermodynamic equilibrium, the following equation can be obtained.

H$^+$ + NH$_{3(g)}$ ↔ NH$_4^+$ (R1)

The equilibrium constant $H_{NH_3}^*$ is equal to the Henry's constant of NH$_3$ divided by the acid dissociation constant for

NH$_4^+$ (Clegg et al., 1998). $\varepsilon(NH_4^+)$ can be analytically calculated as detailed in Guo et al.(2017a) via the following equation:

$$\varepsilon(NH_4^+) = \frac{[NH_4^+]}{[NH_x]} \cong \frac{\frac{\gamma_{H^+}10^{-pH}}{\gamma_{NH_4^+}}H_{NH_3}^* W_i RT \times 0.987 \times 10^{-14}}{1 + \frac{\gamma_{H^+}10^{-pH}}{\gamma_{NH_4^+}}H_{NH_3}^* W_i RT \times 0.987 \times 10^{-14}}$$ (Eq. 1)

here, $[NH_4^+]$ is the molar concentration of NH$_4^+$ (mole m$^{-3}$). γ is the activity coefficient, which is extracted from the

ISORROPIA II model to account for the non-ideality solution effect. $H_{NH_3}^*$ (atm$^{-1}$) represents the molality-based equilibrium constant, which is T dependent and can be determined using equation (12) in Clegg et al.(1998). W$_i$ (μg m$^{-3}$) is the aerosol water content predicted by ISORROPIA-II. R (J/mole/K) is the universal gas constant. T (K) is ambient temperature and 0.987×10$^{-14}$ is the conversion multiplication factors from atm and μg to SI units.

In Figure 6, $\varepsilon(NH_4^+)$ curve (The "S" shape curve, referred to as "S Curve" hereafter) is plotted against pH based on the mean T (10℃), AWC (100 μg m$^{-3}$), and $\frac{\gamma_{H^+}}{\gamma_{NH_4^+}}$ (2.4) during the haze period. Observation-based $\varepsilon(NH_4^+)$ as a function of pH with varying T and AWC is also shown. Clearly, the observational $\varepsilon(NH_4^+)$ data points are relatively well constrained by the theoretical equation, suggestive of reasonable judgement that $\varepsilon(NH_4^+)$ is controlled by T, AWC, pH, and $\frac{\gamma_{H^+}}{\gamma_{NH_4^+}}$. Under the condition of mean pH (4.6 ± 0.3) during the winter haze period, the

"S curve" derives $\varepsilon(NH_4^+)$ of 0.3, around 3/4 of the mean measured $\varepsilon(NH_4^+)$ (0.4 ± 0.1). Earlier works have also observed higher particle phase fraction than the henry's law constants predicted for water soluble aerosol components (Arellanes et al., 2006;Hennigan et al., 2008;Shen et al., 2018). Another possible factor contributing to the underestimation of $\varepsilon(NH_4^+)$ is the unaccounted effect from organic species, whose role in driving the SNA

formation is thought to be significant (Silvern et al., 2017). The organics have been found to account for 35% of

AWC in the southeast USA (Guo et al., 2015), thus $\varepsilon(NH_4^+)$ would be enhanced by including organic aerosol.

Since the mass concentration of organic aerosol was not available in this study, we did a sensitivity analysis via increasing the AWC by 10, 20 to 90 μg m$^{-3}$ as shown in Figure 7. The pH was not re-calculated using the new AWC

because the co-existed organic aerosol altered pH in a complex way (Battaglia Jr et al., 2019;Wang et al., 2018;Pye et al., 2020). For example, some organic acids increase aerosol acidity thus decrease pH, whereas organic basics (e.g., amines) raise aerosol pH. We found that the best agreement between the predicted and measured $\varepsilon(NH_4^+)$ was achieved when we increase the AWC by roughly 90 μg m$^{-3}$, suggesting a nearly 48% of AWC contributed by the organics. This result falls in the range from a recent report in North China that organics contribute to 30 % ± 22% of

AWC (Jin et al., 2020), and slightly higher than those southeastern United States sites that organic aerosol-related water accounting for about 29 to 39% of total water (Guo et al., 2015) and those in the eastern Mediterranean that about 27.5% of total aerosol water resulted from organics (Bougiatioti et al., 2016). To quantitatively determine which parameter dominates the $\varepsilon(NH_4^+)$, the impact on $\varepsilon(NH_4^+)$ from individual variable (i.e. T, AWC, pH, and

$\frac{\gamma_{H^+}}{\gamma_{NH_4^+}}$) during the haze period in winter is assessed (Figure 8). From a theoretical perspective, the decrease of pH and

T, and increase of AWC and $\frac{\gamma_{H^+}}{\gamma_{NH_4^+}}$ would raise $\varepsilon(NH_4^+)$. For instance, in summertime, the lower $\varepsilon(NH_4^+)$ (Figure 4)

are mainly due to higher T that shift the equilibrium to the gas phase, thus higher $NH_3$ ($40 \pm 8$ μg m$^{-3}$) while lower

$NH_4^+$ was observed. Likewise, in wintertime, the lower T facilitates the residence of $NH_4^+$ in the particle phase than the gas phase ($NH_3$: $20 \pm 4$ μg m$^{-3}$), resulting in higher $\varepsilon(NH_4^+)$.

On the basis of "S curve" (Figure 8), each 0.1 unit change of $\varepsilon(NH_4^+)$ can be caused by approximate 5 ℃, 75 μg m$^{-}$

$^3$, 0.3, and 2 units change of T, AWC, pH, and $\frac{\gamma_{H^+}}{\gamma_{NH_4^+}}$, respectively. Actually, T, pH, and $\frac{\gamma_{H^+}}{\gamma_{NH_4^+}}$ are within a relatively narrow range during the winter haze period (Table 2), suggesting the variation of these parameters shouldn't result in the significant change of $\varepsilon(NH_4^+)$. On the contrary, AWC fluctuates greatly during the study period (Table 2).

Therefore, AWC should be the key factor regulating $\varepsilon(NH_4^+)$. It is well established that AWC is a function of RH

and atmospheric aerosol compositions (Pilinis et al., 1989;Wu et al., 2018;Nguyen et al., 2016;Hodas et al., 2014).

AWC has also been known to promote secondary organic aerosol formation by providing aqueous medium for uptake of reactive gases, gas to particle partitioning, and the subsequent chemical processing (McNeill,

2015;McNeill et al., 2012;Tan et al., 2009;Xu et al., 2017b).

The winter haze pH in this study were ~3 units higher than that of the southeastern United States summer campaign (Nah et al., 2018;Guo et al., 2015;Guo et al., 2017a;Xu et al., 2017a), but close to that of 3.7 in rural Europe (Guo et al., 2018) and 4.2 in North China Plain (Liu et al., 2017), where $NH_3$-rich conditions are prevalent. AWC may act as the major factor, because greater AWC dilute the [H$^+$] and raise the pH. The AWC during the haze period ($82 \pm 105$

μg m$^{-3}$) were much higher than those during the non-haze period ($32 \pm 41$ μg m$^{-3}$).

**3.4 A possible self-amplifying feedback mechanism**

Given that AWC is a function of RH and SNA, a conceptual model of how AWC control $\varepsilon(NH_4^+)$ can be illustrated by a self-amplifying feedback loop (Figure 9). Formation of SNA is initiated by gas-particle conversion of $NH_3$.

Under certain meteorological conditions such as high RH and shallow planetary boundary layer, SNA is subject to uptake moisture and result in the increases of AWC. The enhanced aerosol water dilutes the vapor pressure of semi- volatile species (i.e., nitrate, ammonium and chloride) above the particle and driving semi-volatile species continue to condense (Topping et al., 2013). Based on the discussions above, the increase of AWC would further raise

$\varepsilon(NH_4^+)$, leading to more efficient transformation of $NH_3$ as SNA.

Figure 10 shows the yearly mean diurnal variation of $\varepsilon(NH_4^+)$, AWC, SNA along with T and RH. Apparently, SNA

tracked well with $\varepsilon(NH_4^+)$ and AWC, especially over nighttime. The not well-correlated track between SNA and

AWC and $\varepsilon(NH_4^+)$ during the daytime (8:00-16:00) can be ascribed to the photochemical reactions that lead to

SNA formation. The good correlation between SNA and AWC and $\varepsilon(NH_4^+)$ demonstrated in Figure 10 support the proposed self-amplifying feedback loop in SNA formation.

**4 Conclusion**

Our results demonstrate that $\varepsilon(NH_4^+)$, rather than NH$_3$ concentrations, plays a critical role in driving haze formation in the agricultural NH$_3$ emitted regions. Based on the "S curve" calculation, we have unraveled that AWC is the major factor controlling $\varepsilon(NH_4^+)$. Upon analyzing the cross-correlations between AWC, $\varepsilon(NH_4^+)$ and SNA, we proposed a self-amplifying feedback mechanism of SNA formation that associated with AWC and $\varepsilon(NH_4^+)$. This positive feedback cycle is likely to occur in other rural regions, where high agricultural NH$_3$ emissions are prevalent.

We have shown that high NH$_3$ concentrations may not necessarily lead to strong SNA formation, particularly in the agriculture intensive areas, e.g. the North China Plain (NCP) and the extensive farming lands in Eastern China where the high NH$_3$ levels are still unregulated and increasing (Meng et al., 2018;Warner et al., 2017). Although Liu et al.(2019) have predicted that PM$_{2.5}$ can be slashed by 11-17% when 50% reduction in NH$_3$ from the agricultural sector and 15% mitigation of NO$_x$ and SO$_2$ emissions was achieved, a recent study has demonstrated that only when aerosol pH drops below 3.0, the NH$_3$ reduction would have expected mitigation effects (Guo et al., 2018). The winter haze pH ($4.6 \pm 0.3$) in this study was mostly between 4-5. Our results thus imply that NH$_3$ only may not be an effective solution to tackle air pollution in these regions.

*Data availability.*

The data presented in this paper are available upon request from the corresponding author (huangkan@fudan.edu.cn).

*Author contributions.*

JX and KH conceived the study. JX, JC, and KH performed data analysis and wrote the paper. All authors contributed to the review of the manuscript.

*Competing interests.*

The authors declare that they have no conflict of interest.

*Acknowledgements*

The authors acknowledge support of the National Science Foundation of China (No. 91644105), the National Key

R&D Program of China (2018YFC0213105), and the Natural Science Foundation of Shanghai (19ZR1421100). Jian

Xu   acknowledge   project   funded   by   China   Postdoctoral   Science   Foundation   (2019M651365).

[revised manuscript text omitted]

**Figure 8**: $\varepsilon(NH_4^+)$ as a function of pH during the winter haze period. Other variables are held constant at the mean value during the winter haze period, while varying only the AWC, T, activity coefficients ratio of $\frac{\gamma_{H^+}}{\gamma_{NH_4^+}}$, respectively. Shaded dark areas indicate the winter haze mean pH together with one standard deviation (± 1σ). The areas between the green and red line represent the curve corresponding to mean ± 1σ, note that for AWC mean - 1σ yield a negative value, thus the minimum mass concentration (3 μg m⁻³) was used.

[Figure]

**Figure 9: Schematic of self-amplifying feedback loop for SNA formation.**

[Figure]

**Figure 10: Annual mean diurnal pattern of** $\varepsilon(NH_4^+)$**, AWC, SNA, T, and RH.**

**Table 1. Statistical summary on mass concentrations of PM$_{2.5}$ species and NH$_3$.**

| Unit: μg m$^{-3}$ | PM$_{2.5}$ | SO$_4^{2-}$ | NO$_3^-$ | Cl$^-$ | NH$_4^+$ | NH$_3$ |
|---|---|---|---|---|---|---|
| non-haze | 28.5 ± 16.9 | 5.6 ± 3.6 | 6.9 ± 6.6 | 1.1 ± 0.9 | 5.6 ± 3.3 | 32.2 ± 11.6 |
| haze | 98.3 ± 37.2 | 13.3 ± 7.7 | 23.1 ± 14.5 | 2.2 ± 1.9 | 13.2 ± 6.6 | 32.3 ± 13.5 |

**Table 2: The summer and winter mean (±1σ) ε($NH_4^+$), pH, T, activity coefficients ratio of $\frac{\gamma_{H^+}}{\gamma_{NH_4^+}}$, and NH₃ (µg m⁻³) during the haze period.**

| | ε($NH_4^+$) | AWC | pH | NH₃ | $\frac{\gamma_{H^+}}{\gamma_{NH_4^+}}$ | T |
|---|---|---|---|---|---|---|
| summer | $0.2 \pm 0.1$ | $79 \pm 73$ | $3.4 \pm 0.5$ | $40 \pm 8$ | $1.9 \pm 0.9$ | $29 \pm 5$ |
| winter | $0.4 \pm 0.1$ | $115 \pm 131$ | $4.6 \pm 0.3$ | $20 \pm 4$ | $4.0 \pm 4.7$ | $5 \pm 4$ |